# Mallampati Score Is an Independent Predictor of Active Oxygen Therapy in Patients with COVID-19

**DOI:** 10.3390/jcm11112958

**Published:** 2022-05-24

**Authors:** Maciej Dyrbuś, Aleksandra Oraczewska, Szymon Szmigiel, Szymon Gawęda, Paulina Kluszczyk, Tomasz Cyzowski, Marek Jędrzejek, Paweł Dubik, Michał Kozłowski, Sebastian Kwiatek, Beata Celińska, Michał Wita, Ewa Trejnowska, Andrzej Swinarew, Tomasz Darocha, Adam Barczyk, Szymon Skoczyński

**Affiliations:** 1Pyrzowice Temporary Hospital, Leszek Giec Upper-Silesian Medical Center, 40-635 Katowice, Poland; oraczewskaaleksandra@gmail.com (A.O.); tomaszcyzowski@wp.pl (T.C.); jedrzejekmarek@gmail.com (M.J.); mkozlowski303@gmail.com (M.K.); tomekdarocha@wp.pl (T.D.); simon.mds@poczta.fm (S.S.); 23rd Department of Cardiology, School of Medical Sciences in Zabrze, Medical University of Silesia, 41-800 Zabrze, Poland; 3Department of Pneumonology, School of Medicine in Katowice, Medical University of Silesia, 40-752 Katowice, Poland; abarczyk@sum.edu.pl; 4Student Scientific Society, Department of Pneumonology, School of Medicine in Katowice, Medical University of Silesia, 40-752 Katowice, Poland; szymonszmigiel97@gmail.com (S.S.); szyme.ga@gmail.com (S.G.); kluszczyk.paulina@gmail.com (P.K.); 5Department of Anaesthesiology and Intensive Therapy, Medical University of Silesia, 40-752 Katowice, Poland; 6Department of Cardiology and Structural Heart Diseases, Medical University of Silesia, 40-752 Katowice, Poland; 7Department of Anesthesiology and Intensive Therapy, Hospital of the Ministry of the Interior and Administration, 40-061 Katowice, Poland; paweldubik@wp.pl; 8Division of Internal Diseases Oncology, Gastroenterology, Angiology, Department of Cardiology Intensive Care, Hospital of the Ministry of the Interior and Administration, 40-061 Katowice, Poland; sebastiankwiatek@tlen.pl; 9Consultant in Infectious Diseases GCM, Upper Silesian Medical Center, 40-635 Katowice, Poland; bpuzanowska@poczta.onet.pl; 10First Chair and Department of Cardiology, Faculty of Medical Sciences in Katowice, Medical University of Silesia, 40-752 Katowice, Poland; michalwita21@gmail.com; 11Department of Cardiac Anaesthesia and Intensive Care, Silesian Centre for Heart Diseases in Zabrze, Faculty of Medical Sciences in Zabrze, Medical University of Silesia, 40-752 Katowice, Poland; ewatrejnowska@gmail.com; 12Faculty of Science and Technology, University of Silesia in Katowice, 41-500 Chorzów, Poland; andrzej.swinarew@gmail.com; 13Department of Swimming and Water Rescue, Institute of Sport Science, The Jerzy Kukuczka Academy of Physical Education, 40-065 Katowice, Poland

**Keywords:** COVID-19, high-flow nasal cannula, Mallampati score, mortality, non-invasive mechanical ventilation, respiratory failure

## Abstract

Mallampati score has been identified and accepted worldwide as an independent predictor of difficult intubation and obstructive sleep apnea. We aimed to determine whether Mallampati score assessed on the first patient medical assessment allowed us to stratify the risk of worsening of conditions in patients hospitalized due to COVID-19. A total of 493 consecutive patients admitted between 13 November 2021 and 2 January 2022 to the temporary hospital in Pyrzowice were included in the analysis. The clinical data, chest CT scan, and major, clinically relevant laboratory parameters were assessed by patient-treating physicians, whereas the Mallampati score was assessed on admission by investigators blinded to further treatment. The primary endpoints were necessity of active oxygen therapy (AOT) during hospitalization and 60-day all-cause mortality. Of 493 patients included in the analysis, 69 (14.0%) were in Mallampati I, 57 (11.6%) were in Mallampati II, 78 (15.8%) were in Mallampati III, and 288 (58.9%) were in Mallampati IV. There were no differences in the baseline characteristics between the groups, except the prevalence of chronic kidney disease (*p* = 0.046). Patients with Mallampati IV were at the highest risk of AOT during the hospitalization (33.0%) and the highest risk of death due to any cause at 60 days (35.0%), which significantly differed from other scores (*p* = 0.005 and *p* = 0.03, respectively). Mallampati IV was identified as an independent predictor of need for AOT (OR 3.089, 95% confidence interval 1.65–5.77, *p* < 0.001) but not of all-cause mortality at 60 days. In conclusion, Mallampati IV was identified as an independent predictor of AOT during hospitalization. Mallampati score can serve as a prehospital tool allowing to identify patients at higher need for AOT.

## 1. Introduction

The COVID-19 pandemic has resulted in the infection of over 430 million people worldwide, and the global death rate of more than 6 million people [1]. It was estimated that approximately 2% of the overall population with SARS-CoV-2 infection would eventually require hospitalization and oxygen therapy [2]. In recent months, the predictors of worse outcomes, such as age and the presence of severe chronic comorbidities, including cardiovascular and pulmonary diseases, were defined [3,4,5,6]. Biomarkers, such as neutrophile-to-lymphocyte ratio and interleukin-6, were established as independent predictors of COVID-19 worsening, although their evaluation was not possible until the blood sample was taken [7,8].

In the 1980s, the Mallampati score was identified to independently predict difficulty in intubation in patients before surgery, as well as to foresee the presence of obstructive sleep apnea [9,10]. The score which consists of four grades considers the mutual relationship of the uvula, throat, and soft palate and allows one to visually characterize the anatomical structures of the upper respiratory tract (URT). As patients with COVID-19 are prone to respiratory failure due to pneumonia, the relationship of anatomic structures constituting the URT might contribute to worsening ventilation mechanics and increasing necessity of active oxygen therapy (AOT) introduced to support the failing respiratory system.

The purpose of the analysis was to investigate the influence of the URT anatomy assessed with the use of the Mallampati score on the outcomes of patients hospitalized due to COVID-19 in the large facility dedicated to COVID-19 patients.

## 2. Materials and Methods

### 2.1. The Population

The consecutive patients admitted to the temporary hospital in Pyrzowice between 13 November 2021 and 2 January 2022 were included in the analysis. In those patients, the demographic and clinical data, such as the patient’s medical history, baseline laboratory parameters, baseline chest CT scan, and the Mallampati score, were assessed by the investigators on the day of admission and included in the database. Obesity was defined as body mass index (BMI) of 30.0 kg/m^2^ or higher. The physicians-in-charge were blinded to the results of the Mallampati score.

### 2.2. The Facility

The facility has been providing continuous care since March 2021 and has been designed to accommodate 134 patients in the internal medicine unit’s beds fully equipped to provide oxygen therapy, including high-flow nasal oxygen therapy (HFNO) and noninvasive mechanical ventilation (NIV), and 14 patients in the intensive care unit (ICU). The unit was devoted to accommodate patients with respiratory failure due to COVID-19 in case of lack of available hospital beds. Patients with trauma, multiorgan failure, or patients who were SARS-CoV-2-positive requiring admission due to surgical causes were not admitted to the hospital and not included in the analysis.

Each patient admitted to the hospital was required to have a chest CT scan, if such an exam had not been performed within 24 h before admission. The pharmacotherapy was based on current guidelines against SARS-CoV-2, including remdesivir, dexamethasone, tocilizumab, and baricitinib [11]. The choice of the type of active or passive oxygen therapy, as well as all other clinical decisions, were based solely on the patient’s clinical presentation assessed by the patient’s physician-in-charge, with HFNO and NIV available for the escalation of oxygen therapy if needed.

The intensive care unit has provided care almost exclusively for patients hospitalized in the internal medicine ward, as well as for some patients who were transferred from the intensive care unit to the intensive care facilities in the municipal hospitals in the vicinity of the temporary hospital.

### 2.3. The Study Outcomes

The primary outcomes of the study were the necessity of the escalation of oxygen therapy to active oxygen therapy, which consisted of NIV or HFNO, during the hospitalization, and the all-cause death at 60 days from admission. The secondary outcomes were the need for escalation of oxygen therapy to either of the active therapy modalities, as well as the need for transfer to the intensive care unit, and in-hospital mortality due to any cause.

### 2.4. The Statistical Analyses

The categorical variables are presented using frequency tables for both absolute numbers and percentages. The continuous variables are summarized using arithmetic mean with standard deviation for data following a normal distribution or median with a quartile 1 and 3 for data demonstrating a non-normal distribution.

Normality of distribution was tested using the Shapiro–Wilk test. Between-group comparisons of continuous variables were conducted using ANOVA (if normally distributed); otherwise, the Kruskal–Wallis test was used. The Pearson’s chi-squared test was used to evaluate categorical variables. The interval of two-sided *p* < 0.05 was considered statistically significant. Unifactorial and multifactorial analyses were performed to assess the variables using the logistic stepwise backward regression model (*p* < 0.05 for inclusion in the model, *p* < 0.05 for remaining in the model). All investigated statistically significant clinical and laboratory parameters were included in the unifactorial analysis after the exclusion of co-dependent variables in the correlation analysis. Estimated parameter values are presented as odds ratios (ORs) with a 95% confidence interval (CI). STATISTICA 10 (StarSoft Inc., Tulsa, OK, USA) was used for all calculations. The approval of a bioethics committee was not required, based on the PCN/CBN/0022/KB/263/21 decision of the Bioethics Committee of the Medical University of Silesia.

## 3. Results

During the analyzed period, a total of 599 patients were admitted and treated in the facility, of whom in 493, the detailed characteristics were available for the analysis. Among them, 69 (14.0%) patients were in Mallampati I, 57 (11.6%) in Mallampati II, and 78 (15.8%) in Mallampati III, while the majority of patients (289; 58.6%) were classified as Mallampati IV. The detailed demographic and clinical characteristics are presented in Table 1.

There were 44.8% of female patients, and the median age of the overall population was 69 years. The population was burdened with risk factors, including the presence of hypertension (56.2%), chronic kidney disease (26.0%), diabetes (24.5%), and obesity (26.7%) diagnosed before hospital admission. There were (31.2%) patients who had been fully vaccinated on admission. The presence of comorbidities did not differ significantly concerning the Mallampati score, with the exception of chronic kidney disease, which was more present in patients classified as Mallampati IV (30.5%) than in the remaining classes. A trend toward a higher prevalence of diabetes could be observed in patients with Mallampati IV score. The baseline laboratory results, as well as the patients’ percentage of pneumonia involvement, are presented in Table 2.

There were no differences in any of the parameters concerning the analyzed Mallampati score, and the average lung involvement was 30% (10–50%), regardless of the Mallampati score.

During the hospitalization, 27.0% of patients required an escalation of oxygen therapy to active oxygen therapy as presented in Table 3.

There were significantly more patients in Mallampati IV who required either HFNO or NIV during hospitalization, with the majority of those patients requiring NIV, not HFNO, as a destination therapy, as seen in Figure 1.

Almost one-third of patients in Mallampati IV score required AOT, in contrast to 17.4% and 17.5% from the first and second Mallampati score, respectively. In the multivariable analysis, Mallampati IV was found as an independent predictor of active oxygen therapy (OR: 3.089, 95% CI 1.654–5.770), but not of all-cause mortality at 60 days, as presented in Figure 2.

## 4. Discussion

According to our best knowledge, we were able to demonstrate for the first time that: (1) patients with Mallampati IV were at higher risk of escalation to active oxygen therapy, and (2) the two-month mortality in patients with COVID-19 treated in the large, dedicated facility was 31.2% and was higher in patients classified as Mallampati III or IV.

Clinically, patients with non-SARS-CoV-2-related pneumonia should usually undergo risk stratification based on the established risk scores, such as the CRB-65 score [12,13]. The other scores stratifying the risk in patients with pneumonia, including SCAP, PSI/PORT, or CURB-65 scores, are based on a broader spectrum of parameters; however, they require blood sample analysis, and thus they cannot be performed in the outpatient, prehospital conditions [14,15]. It should be noted that the clinical course of COVID-19 pneumonia is often rapid and unpredictable, with the majority of patients requiring oxygen therapy prior to admission. Thus, the CRB-65 score was not evaluated in our analysis. Taking into account the frequent problem in an appropriate triage of COVID-19 patients, it has been found that patients frequently presented better on clinical assessment and then deteriorated rapidly [16,17]. Therefore, we searched for a clinical parameter that is unchangeable by respiratory rate, oxygen supplementation, respiratory muscle strength, patient responsiveness, or age.

The COVID-19 pandemic has urged a wide adoption of active oxygen therapy in response to acutely deteriorating ventilation capacity. Prior to COVID-19, its use has been less prolific, and mostly restricted to acute hypoxemic, normocapnic respiratory failure being a result of various causes. In other studies, different indications, such as an immunocompromised population, pre-intubation, or trauma-setting, have been defined. The majority of the studies, which have investigated AOT efficacy and safety in acute respiratory failure, have utilized the inclusion criteria based on PaO_2_/FiO_2_ quotient or arterial blood saturation [18,19,20]. A large meta-analysis of acute hypoxemic respiratory failure development de novo included patients who had at least one of the following respiratory failure manifestations: PaO_2_ /FIO_2_ ≤ 300, PaO_2_ ≤ 65 mm Hg, or SpO_2_ ≤ 92% with signs and symptoms of respiratory distress [21]. The results indicated that the use of NIV solely reduced the rate of intubations, although no AOT modalities affected long-term outcomes. Nonetheless, the authors of ERS/ATS guidelines on the use of non-invasive ventilation in acute respiratory failure from 2017 did not clearly recommend the use of NIV or HFNO in patients with de novo ARF, despite the survival benefit derived from the pooled study analyses, which could be explained by the low certainty of the results of which the evidence consisted [22].

As COVID-19 pneumonia causes significant deterioration in patients’ respiratory conditions, the percentage of patients who required escalation of oxygen therapy to the active therapy was higher than prior to COVID-19. In the era of COVID-19, the reports claim that HFNO has been used in 23–64% of patients with severe COVID-19 pneumonia [23]. Data on HFNO in COVID-19 indicate that its use has reduced the necessity of endotracheal intubation by between 44% and as much as 64% but has had no clear effect on mortality [23]. The subanalysis of the HOPE COVID-19 registry indicated that more than half of the patients who received NIV due to COVID-19 pneumonia survived without intubation [24]. Nonetheless, for the group of patients in whom NIV treatment led to clinical failure, the in-hospital mortality reached 58%, being significantly higher compared with the group in which NIV led to respiratory success.

Until September 2021, no clear guidelines summarizing the approach to oxygen therapy in patients with COVID-19 had been presented due to rapid progression of the disease and the need to react quickly to respiratory distress. In the review article by Akoumianaki et al., the proposed scheme of therapy escalation was NIV/HFNO introduction if SpO_2_ was lower or equal to 90% on the conventional oxygen therapy with 6–12 L/min flow, with an emphasis on earlier HFNO than NIV introduction, mostly due to its better tolerability [23]. According to Ref. [25], as mentioned above, in September 2021, the ERS published guidelines on the use of HFNO in ARF, including COVID-19 [26]. The authors recommend the use of HFNO over NIV in patients with progressive or moderate to severe ARF, owing mostly due to the evidence suggesting lower rate of intubation, and potentially lower risk of death in patients ventilated with HFNO. However, the use of HFNO is rather unchangeable, as it allows modification of three respiratory parameters (air temperature, oxygen quotient, and flow), while the use of NIV might differ in terms of ventilation mode (spontaneous continuous positive airway pressure—CPAP or controlled biphasic positive airway pressure—BIPAP), the pressures and volumes set for specific patients, the percentage of oxygen in the inspiratory air, as well as the duration of NIV treatment throughout the day. Furthermore, to compensate for hypoxemia, higher FiO_2_ is required when comparing HFNO to NIV; however, high oxygen concentration might bring a similarly devastating effect on alveoli as COVID-19 [25,27]. Moreover, higher positive airway pressure in NIV (usually 8–14 cmH_2_O) when compared with HFNO (estimated at 5 cmH_2_O) may result in more effective alveolar recruitment. In all treated patients, awake proning was used whenever tolerated by patients [25,26].

In our practice, the NIV was mostly utilized in patients with clinical signs of respiratory fatigue and in hemodynamically susceptible patients, such as those with congestive heart failure. Moreover, as the muscle tone, including the respiratory muscles, relaxes during sleep, the vast majority of our patients spent at least 12 h daily on the NIV ventilation, owing to our preference to stabilize the airways and prevent hypoventilation, when the patients were asleep and the muscular tone relaxed [28]

Taking into consideration that the overall median pneumonia involvement was 30%, it was not surprising that 27% of our patients required either HFNO or NIV and the percentage of patients being treated with NIV as the destination therapy was higher than with HFNO (18.4% vs. 8.7%). The patients from higher Mallampati grades had a higher need for active oxygen therapy, with a clear trend to introduce NIV in these patients, while Mallampati IV was identified as an independent predictor of active oxygen therapy in our analysis.

The physiological rationale supporting our results is that the Mallampati score evaluates the anatomical relationship of the structures building URT, indicating a higher risk of airway obstruction and restriction in flow through the URT. Patients with a restricted capacity of URT at baseline seemed to be at higher risk of hypoventilation and progression of hypoxemic respiratory failure, which would eventually require HFNO or NIV. The higher percentage of NIV demonstrated the necessity to increase the positive airway pressure in URT while acting as a respiratory “pneumatic splint”. Therefore, in those patients, NIV acted bidirectionally, recruiting pulmonary alveoli and stabilizing the upper respiratory tract.

It has been confirmed that the presence of comorbidities increases the risk of death due to COVID-19 [3,5,6,29,30]. In our analysis, the clinical predictors of all-cause death were age, which has already been established as one of the most crucial factors facilitating survival in COVID-19, and chronic kidney disease [31,32]. Patients with impaired renal function are not only prone to volume imbalance but also to a higher risk of concomitant bacterial infections, and often require treatment modifications due to impaired filtration [33,34]. Moreover, some drugs which have shown benefit in treatment of COVID-19, including remdesivir and tocilizumab, are contraindicated in severe chronic kidney disease; thus, the anti-COVID treatment strategies in these patients are also restricted.

On the other hand, the pneumonia volume did not differ significantly between the Mallampati scores, which suggests that no interplay between the upper respiratory tract anatomy and percentage of diseased lungs was identified in our study. Mallampati score was not stated as an independent predictor of outcomes in our population, although all-cause mortality in 60 days was higher in patients with Mallampati III and IV. We believe that such result could be explained by a slightly worse clinical profile of those patients, who had a higher prevalence of chronic kidney disease and a trend for a higher occurrence of diabetes.

Additionally, an important factor influencing the high mortality rate of studied patients was a small percentage of full vaccination, which was 31.2% in the overall studied cohort. Although in our analysis, vaccination was not found to independently increase the risk of either active oxygen therapy or mortality, it may be due to the relatively small studied population and the presence of other confounders, especially since contemporary large studies indicate a significant impact of lack of full vaccination on worse prognosis and higher mortality in patients with COVID-19 [35,36].

Finally, laboratory biomarkers, such as elevated ferritin or interleukin-6 levels, have already been identified as risk factors for a significantly worse prognosis in COVID-19 patients [37,38,39,40]. However, no such result was demonstrated in our analysis—the sole laboratory parameter independently associated with higher all-cause mortality was white blood count.

The COVID-19 pandemic has altered the functioning of the healthcare systems across the globe, with the necessity to provide care for patients with COVID-19, as well as to accommodate the growing needs of patients with non-COVID-related admissions. We suggest the Mallampati score assessment, which was described in our study as an easy and useful tool that can hypothetically be performed by trained paramedics in the prehospital setting. Such evaluation can serve as a risk-stratifying tool to study patients at higher need for active oxygen therapy and provide clinical guidance. It may benefit the facilities to provide such therapy for patients at higher risk of requiring AOT instead of prolonging the treatment with conventional oxygen therapy, especially since the ventilatory capacity of patients with COVID-19 pneumonia often deteriorates dramatically and rapidly. It can also be hypothesized that the addition of the Mallampati score may more properly guide clinical decision making in patients monitored at home, since it has served as a promising tool to reduce hospitalizations in patients with COVID-19 (43,44), which consequently may also decrease the healthcare system costs.

### 4.1. Limitations

The analysis performed by our team possesses certain limitations one has to acknowledge. First, although data were gathered prospectively, the results were conducted in a retrospective fashion; thus, the causality of interactions could not be ascertained. Secondly, the detailed characteristics of 493 of 599 patients were available for analysis, while in the remaining patients, the Mallampati score could not be analyzed, either due to their urgent admission to the intensive care unit, the necessity to quickly introduce invasive ventilation, or other reasons. Third, the decision to obtain arterial blood gas analysis on admission was solely at the discretion of the physician-in-charge; therefore, in the majority of patients, the results of arterial blood gas analysis were not available for the purpose of this analysis. Nevertheless, it should be noted that in patients in whom the arterial blood gas analysis was performed, especially on the days of massive admissions, and in patients with worse clinical presentation, the passive oxygen treatment could have been implemented before arterial blood gas assessment was performed. Thus, the generalization of the results could have been significantly biased by the passive oxygen supply from the emergency medical services. Finally, the follow-up analysis was performed based on data from the electronic databases of the National Health Fund (*Narodowy Fundusz Zdrowia*—*NFZ*).

### 4.2. Areas for Future Research

Based on our findings, it can be speculated that the Mallampati score might have a potential role for prehospital patients’ triage and assessment. To reveal this, our trial should be performed directly in a prehospital setting, where the decision to leave the patient at home, transfer to the closest hospital, or transfer to a hospital equipped with AOT is based also on the Mallampati score assessed by paramedics or general practitioners (GPs).

### 4.3. Interpretation

Our results demonstrated that in a large cohort of patients with COVID-19 pneumonia, the Mallampati score assessed on admission could identify those at higher risk of requiring active oxygen therapy. Mallampati IV was identified as an independent predictor of active oxygen therapy during hospitalization, but not of increased 60-day mortality.

## Figures and Tables

**Figure 1 jcm-11-02958-f001:**
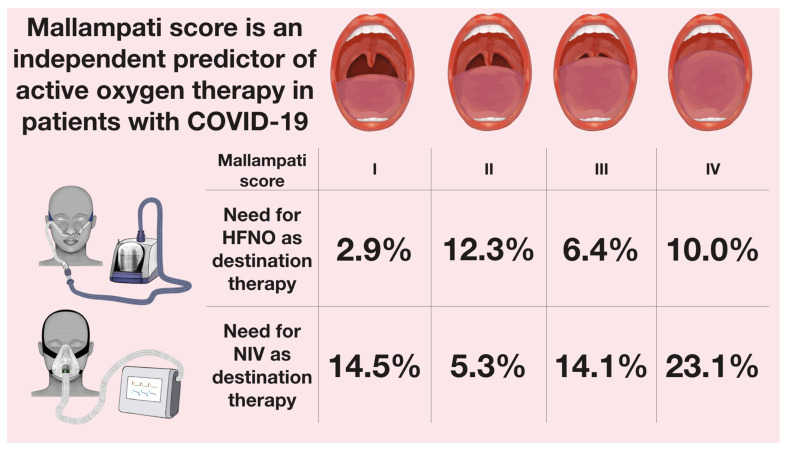
The percentage of patients with COVID-19 who required active oxygen therapy depending on the Mallampati score. Abbreviations: HFNO—high-flow nasal oxygen therapy; NIV—non-invasive ventilation.

**Figure 2 jcm-11-02958-f002:**
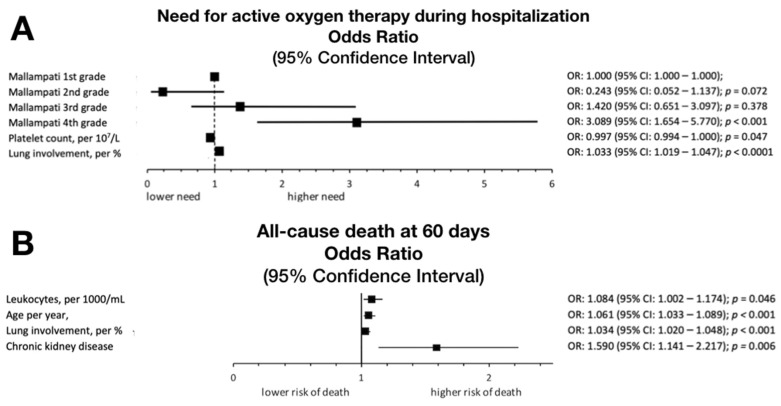
Multivariable analysis indicating independent predictors and the influence of consecutive Mallampati grades on the need for active oxygen therapy during hospitalization (**A**) and independent predictors of all-cause mortality at 60 days (**B**).

**Table 1 jcm-11-02958-t001:** Demographic and clinical characteristics at baseline.

Parameter	Overall	Mallampati I	Mallampati II	Mallampati III	Mallampati IV	*p*
Female sex, n/N (%)	220/493 (44.8%)	26/69 (37.7%)	27/57 (47.4%)	36/78 (46.2%)	131/289 (45.4%)	0.64
Age, years	69 (61–79)	65 (55–77)	67 (58–73)	69 (63–78)	69 (61–77)	0.29
CKD	128/493 (26.0%)	14/69 (20.3%)	9/57 (15.8%)	17/78 (21.8%)	88/289 (30.5%)	0.046
Asthma	40/493 (8.1%)	6/69 (8.7%)	3/57 (5.3%)	3/78 (3.9%)	28/289 (9.7%)	0.32
COPD	31/493 (6.3%)	2/69 (2.9%)	5/57 (8.8%)	5/78 (6.4%)	19/289 (6.6%)	0.57
OSA	14/493 (2.9%)	1/69 (1.5%)	0/57 (0.0%)	2/78 (2.6%)	11/289 (3.8%)	0.37
DM	121/493 (24.5%)	12/69 (17.4%)	11/57 (19.3%)	15/78 (19.2%)	83/289 (28.7%)	0.08
HA	277/493 (56.2%)	46/69 (66.7%)	29/57 (50.9%)	39/78 (50.0%)	163/289 (56.4%)	0.18
CAD	131/493 (26.6%)	16/69 (23.2%)	11/57 (19.3%)	19/78 (24.4%)	85/289 (29.4%)	0.34
Stroke	47/493 (9.5%)	5/69 (7.2%)	3/57 (5.3%)	6/78 (7.7%)	33/289 (11.4%)	0.37
Malignancy	59/493 (12.0%)	7/69 (10.1%)	8/57 (14.0%)	9/78 (11.5%)	35/289 (12.1%)	0.98
Smoking	76/493 (15.4%)	5/69 (7.2%)	12/57 (21.1%)	14/78 (18.0%)	45/289 (15.6%)	0.14
Obesity	130/493 (26.3%)	14/69 (20.3%)	16/57 (28.1%)	22/78 (28.2%)	78/289 (26.9%)	0.66
Full vaccination	154/493 (31.2%)	23/69 (33.3%)	15/57 (26.3%)	27/78 (34.6%)	89/289 (30.8%)	0.80

Abbreviations: CAD—coronary artery disease; CKD—chronic kidney disease; COPD—chronic obstructive pulmonary disease; DM—diabetes mellitus; HA—hypertension; and OSA—obstructive sleep apnea.

**Table 2 jcm-11-02958-t002:** Laboratory and imaging parameters at baseline.

Parameter	Overall	Mallampati I	Mallampati II	Mallampati III	Mallampati IV	*p*
BMI, kg/m^2^ (Q1–Q3)	27.5 (24.6–30.9)	26.4 (24.2–29.4)	27.3 (24.2–31.7)	27.0 (24.3–30.5)	27.8 (25.0–31.1)	0.20
Pneumonia volume, %	30 (10–50)	20 (10–40)	20 (15–35)	25 (10–50)	30 (10–50)	0.23
Platelets; median (Q1–Q3)	195 (147–273)	196 (148–270)	196 (153–313)	192 (153–277)	200 (151–276)	0.87
Hemoglobin median (Q1–Q3)	13.8 (12.3–16.1)	14.2 (13.2–16.4)	13.3 (11.9–16.0)	13.5 (12.2–15.9)	13.7 (12.2–15.4)	0.49
WBC median (Q1–Q3)	6.5 (4.7–9.1)	6.0 (4.1–8.9)	5.5 (4.4–7.6)	6.4 (4.6–8.8)	6.7 (4.8–9.0)	0.1
CRP median (Q1–Q3)	89 (50–146)	65 (36–114)	90 (59–131)	95 (54–133)	86 (48–147)	0.19
PCT median (Q1–Q3)	0.14 (0.07–0.31)	0.09 (0.05–0.24)	0.11 (0.06–0.26)	0.13 (0.06–0.27)	0.13 (0.07–0.3)	0.34
IL-6 median (Q1–Q3)	46.9 (21.3–92.0)	34.7 (15.9–80.2)	42.5 (22.1–79.2)	38.6 (17.2–65.8)	48.3 (21.3–95.6)	0.14
D-Dimer median (Q1–Q3)	1160 (670–2120)	845 (492–1955)	1035 (695–2075)	945 (640–2100)	1160 (670–2000)	0.45
Pulse oximeter oxygen saturation, %, median (Q1–Q3); [n/N] *	88 (83–93) [279/494]	90 (85–95) [44/69]	90 (85–94) [36/57]	88 (84–93) [48/78]	88 (82–93) [151/289]	0.14

Abbreviations: BMI—body mass index; CRP—C-reactive protein; PCT—procalcitonin; WBC—white blood count; and *—arterial saturation derived from pulse oximetry, regardless of the use of oxygen therapy in the emergency medical services.

**Table 3 jcm-11-02958-t003:** Outcomes in patients with COVID-19 based on Mallampati score assessment.

Parameter	Overall	Mallampati I	Mallampati II	Mallampati III	Mallampati IV	*p*
Transfer to ICU, n/N (%)	66/493 (13.4%)	11/69 (15.9%)	3/57 (5.3%)	7/78 (9.0%)	45/289 (15.7%)	0.10
In-hospital death n/N (%)	100/493 (20.3%)	13/69 (18.8%)	7/57 (12.3%)	17/78 (21.8%)	63/289 (21.8%)	0.40
PE during hospitalization; n/N (%)	33/493 (6.7%)	3/69 (4.3%)	1/57 (1.8%)	5/77 (6.4%)	24/289 (8.3%)	0.27
Active oxygen therapy; n/N (%)	133/493 (27.0%)	12/69 (17.4%)	10/57 (17.5%)	16/78 (20.5%)	95/289 (32.9%)	0.005
HFNO as destination therapy	43/493 (8.7%)	2/69 (2.9%)	7/57 (12.3%)	5/78 (6.4%)	29/289 (10.0%)	0.024
NIV as destination therapy	91/493 (18.5%)	10/69 (14.5%)	3/57 (5.3%)	11/78 (14.1%)	67/289 (23.2%)	
Days to HFNO	1 (0–3)	1 (0–3)	2 (1–3)	2 (1–4)	1 (0–3)	0.8
Days to NIV	1 (0–3)	4 (2–5)	1 (1–1)	1.5 (1–3.5)	2 (1–4)	0.53
Death at median of 60 days n/N (%)	154/493 (31.2%)	19/69 (27.5%)	9/57 (15.8%)	25/78 (32.1%)	101/289 (35.0%)	0.03

Abbreviations: HFNO—high-flow nasal oxygen therapy; ICU—intensive care unit; NIV—non-invasive ventilation; and PE—pulmonary embolism.

## Data Availability

All source data can be provided on the request of the reader.

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
