# Peer review of "Mallampati Score Is an Independent Predictor of Active Oxygen Therapy in Patients with COVID-19"

_jcm, 2022, doi:10.3390/jcm11112958_

Round 1

Reviewer 1 Report

Thank you for the opportunity to review this manuscript by Maciej DyrbuÅ› and colleagues titled  "Mallampati score is an independent predictor of active oxygen therapy in patients with COVID-19 ". The study is well written and interesting however no data of arterial blood gas analysis of the patients at the arrival in hospital are shown in the manuscript. Indeed it would be interesting to know if there was any association with it( paO2 or PF for instance)

It seems that mallampati 3 and 4 patients are also overweight or obese which is kind of expected however no correlation with overweight o obesity is underlined or discussed in the paper. Was obesity considered in the multivariate analysis of predictors of need for active oxygen therapy during hospitalization?

Moreover, the percentage of fully vaccinated patients is very low which per se can predispose to severe disease especially in patients with comorbidities. please comment

In the limitiations I would avoid mentioning that "Another limitation is a theoretically possible increased risk of viral transmission of health care workers, exposed to patients’ aerosol containing SARS-CoV2 particles. However, the proper use of personal protective  equipment significantly reduces the risk of infection of health care workers (43,44)" which does not really represent a limitation and it may be discussed but not included in this paragraph .

thank you

Author Response

Remark #1:

"Thank you for the opportunity to review this manuscript by Maciej DyrbuÅ› and colleagues titled  "Mallampati score is an independent predictor of active oxygen therapy in patients with COVID-19 ". The study is well written and interesting however no data of arterial blood gas analysis of the patients at the arrival in hospital are shown in the manuscript. Indeed it would be interesting to know if there was any association with it (paO2 or PF for instance)

Response #1:
Dear reviewer, we would like to agree that the data regarding
arterial gas blood test performed on admission would be of great interest. However, only some patients had arterial blood gas analysis performed on admission, and in most cases, such result was used solely for the further therapy guidance for the physician in-charge, and was not available for the purpose of this analysis.
Nonetheless, the patients admitted to our facility had their oxygen saturation measured using pulse oximetry, and this result was available for the analysis in 279 patients, thus constituting 55% of the studied population. The results indicate that the median SpO2 was 88%, and there was only a slight trend towards lower arterial oxygen saturation in patients with higher Mallampati scores. Moreover, the multivariable analysis did not identify oxygen saturation on admission as an independent predictor of worse prognosis.

Nevertheless, it should be noted that in patients in whom the arterial blood gas analysis was performed, it has been made directly at the emergency room, whereas especially on the days of massive admissions (the recruitment to the analysis took place at the 4th wave of the pandemic, with often more than 20 patients admitted daily to the facility) the passive oxygen treatment has been implemented in significant number of patients before arterial blood gas assessment could have been performed, and thus, the generalization of the results could have been significantly biased by the passive oxygen supply from the emergency medical services.

Parameter

Overall

Mallampati I

Mallampati II

Mallampati III

Mallampati IV

P

Baseline arterial oxygen saturation, %, median (Q1-Q3); [n/N]

88 (83-93) [279/494]

90 (85-95) [44/69]

90 (85-94) [36/57]

88 (84-93) [48/78]

88 (82-93) [151/289]

0.14

Remark #2:
It seems that mallampati 3 and 4 patients are also overweight or obese which is kind of expected however no correlation with overweight o obesity is underlined or discussed in the paper. Was obesity considered in the multivariate analysis of predictors of need for active oxygen therapy during hospitalization?

Response #2:
We would like to thank the Reviewer for presenting interest in the characteristics of our population. Indeed, there is a tendency for higher prevalence of obesity in patients with higher Mallampati score, as the obese patients are at a naturally higher risk of upper respiratory tract obstruction. However, although we have seen a subtle trend for higher BMI between patients from the 4th and the 1st quartile, no significance between the subgroups was found.
However, it should be noted that the the median BMI of our population has been 27.5, which places the majority of our overall population in the overweight cohort, especially when one observes that for BMI, the 25th percentile of the whole population was 24.6, which indicates that only approximately 25% of all patients had BMI in the normal range. Since no differences have been found between subgroups with regard to BMI, we did not intend to focus the attention of the Reader on the further confounders, especially since it was Mallampati, not BMI, which has been identified as an independent predictor of worse outcome in our population. Finally, in the population of patients included in our study, there were no differences in the prevalence of OSA with regard to the Mallampatii scores, although one should remain cautious, because only minority of patients had been assessed with whole night polysomnography before admission, therefore OSA could have been underdiagnosed in this population.

The exact results of the anthropometric analyses in the studied population are presented below. However, we have decided to publish solely the direct result, and the most important of the three parameters being the BMI.

Parameter

Overall

Mallampati I

Mallampati II

Mallampati III

Mallampati IV

P

BMI, kg/m2
(Q1-Q3)

27.5 (24.6-30.9)

26.4 (24.2-29.4)

27.3 (24.2-31.7)

27.0 (24.3-30.5)

27.8 (25.0-31.1)

0.20

Height, cm, median (Q1-Q3)

170 (164-176)

174 (165-178)

170 (164-177)

170 (162-175)

168 (163-176)

Weight, kg, median (Q1-Q3)

80 (70-90)

80 (70-90)

78 (70-90)

78 (69-90)

80 (70-90)

With regard to the multivariable analysis, BMI has been included in the initial univariable analysis, however, it did not meet the prespecified criterion for inclusion in the multivariable model, which as stated in the “Methods” section has been p<0.05. The list of all factors included in the univariable analysis is presented below:

Factors included in the univariable analysis

Baseline demographic factors:
-Mallampati Score
-Time from 1st symptoms to admission
-Gender
-Age
-BMI
-Vaccination status

Laboratory/imaging findings on admission:
-Pneumonia volume (%) on admission
-Platelet count
-Haemoglobin
-White blood count
-Lymphocyte count
-C-reactive protein
-Procalcitonin
-Ferritin
-Interleukin-6
-D-dimer
-Albumin concentration
-Pulse oximeter oxygen saturation

History of:
-Asthma
-COPD
-Obstructive sleep apnea
-Hypertension
-Diabetes
-Coronary artery disease
-Prior stroke
-Prior malignancy
-Smoking (active smoker on admission/history of smoking)

We hope that such response fully explains the question stated by the Reviewer.

Remark #3:
Moreover, the percentage of fully vaccinated patients is very low which per se can predispose to severe disease especially in patients with comorbidities. please comment

Response #3:
We do agree that vaccination against SARS-CoV-2 decreases the risk of severe disease. In the countries with a higher percentage of the fully vaccinated population, there were lower rates of mortality and/or worse outcome in patients infected with SARS-CoV-2. The rate of complete vaccination in Poland during the conduction of the study has been between 53.0% on the 13th of November 2021 (1st day of enrollment) and 55.5% on the 2nd of January, 2022 (the last enrollment day). Thus, the percentage of patients fully vaccinated among patients hospitalized due to severe SARS-CoV-2 infection must have been lower than in overall population and, as presented in our analysis, was 31.2%.
However, there were no differences in the complete vaccination rate with regard to the Mallampati score cluster (p=
0.80) and vaccination has not been demonstrated to be a significant predictor of either active oxygen therapy necessity, or of mortality among our patients.

Remark #4:
In the limitations I would avoid mentioning that "Another limitation is a theoretically possible increased risk of viral transmission of health care workers, exposed to patients’ aerosol containing SARS-CoV2 particles. However, the proper use of personal protective equipment significantly reduces the risk of infection of health care workers (43,44)" which does not really represent a limitation and it may be discussed but not included in this paragraph.

Response #4:
Thank you for this remark. According to the suggestion of the Reviewer we have deleted the mentioned paragraph from manuscript’s discussion, as well as the References to this section.

Reviewer 2 Report

I am thankful for giving me the chance to review the manuscript entitled “Mallampati score is an independent predictor of active oxygen therapy in patients with COVID-19”. This study has some positive findings; however, I have few concerns:

  1. To my knowledge, the proportion of Mallampati IV was not as high as 60% in general population. Do the authors have data to describe the proportion of Mallampati IV in general population in Poland? Or do the authors have data to show the proportion of Mallampati IV was higher in patients with COVID-19.
  2. Maybe, you have some miss data in Table 1 (smoking and obesity) and Table 3. However, the data in Table 3 was not compatible from the original data (ex: 494 > 493, 290 > 289). Please check the data again.
  3. “In the multivariable analysis, Mallampati IV has been found as an independent predictor of active oxygen therapy (OR: 3.089, 95% CI 1.654-5.770), but not of all-cause mortality at 60 days, as presented in Figure 2.” These were the most important findings; however, the authors did not clearly describe the details. (1) Did you mean Mallampati I as baseline for comparison for AOT? If yes, you should also show the result of other grades (II, III). Why did you only show platelet count and lung involvement (not significant data)? How about other parameter? (2) What is the baseline (also Mallampati I?) for comparison for 60-day mortality? What is the data for Mallampati II, III, and IV? Why did you only show significant parameters (leukocytes, age, lung involvement, and CKD)? How about other parameter?

Author Response

Remark #5:
I am thankful for giving me the chance to review the manuscript entitled “Mallampati score is an independent predictor of active oxygen therapy in patients with COVID-19”. This study has some positive findings; however, I have few concerns:
To my knowledge, the proportion of Mallampati IV was not as high as 60% in general population. Do the authors have data to describe the proportion of Mallampati IV in general population in Poland? Or do the authors have data to show the proportion of Mallampati IV was higher in patients with COVID-19.

Response #5:
We would like to thank the Reviewer for this remark. Indeed, we do agree that in the distribution of Mallampati I-IV grades in the Caucasian population, the I-II grades should dominate, although no exact data from Poland, or even from the Central-Eastern Europe are known for the overall population, regardless if analysed with respect to COVID-19 era, or not. Moreover, no data on the distribution of Mallampati grades in patients with SARS-CoV-2 infection are available, what from one perspective allows us to initiate the discussion over a yet undiscussed field of COVID-19, but from the other point of view requires caution as no reference data are available for comparison.

On the other hand, the Mallampati grades tend to positively correlate with obesity, which is another epidemic in developed countries. In our analysis, the 25th percentile of BMI was 24.6, which indicates that approximately three in four patients were overweight, regardless of the Mallampati score. It is likely that obese and overweight patients experienced more pronounced dyspnoea than patients with normal body weight, and thus were more likely to call the emergency services and - therefore –to be admitted to hospital than patients with fewer risk factors.

Finally, we would like to note that obesity has not been identified as a statistically significant predictor in our analysis, and the Mallampati score was assessed and compared by independent individuals, who gathered the data on the body mass and height. The scores were compared and no statistically significant differences with regard to BMI were detected, which suggests that at least in our population, the presence of obesity – in contrary to the Mallampati score -  did not influence outcomes.

Remark #6:

Maybe, you have some miss data in Table 1 (smoking and obesity) and Table 3. However, the data in Table 3 was not compatible from the original data (ex: 494 > 493, 290 > 289). Please check the data again.

Response #6:
We would like to thank the Reviewer for this remark. With regard to the missing data, the number of patients, in whom the information were available for each analysed variable, has been included in the parenthesis. As far as smoking was concerned, there has been a mistake in the number of overall patients, with correct numbers in the respective Mallampati grades.
The obesity has been based on the patients’ medical history, and in 32 patients no information has been obtained. However, to overcome this limitation, we have decided to reclassify obesity and instead of prior assessment based on patients’ history, we have calculated it on the basis of BMI≥30, according to the presently assumed standards. The modified information is now included in the Table 1, as well as in the Methods section, in which the information on the methods of obesity identification was presented stating “
Obesity has been defined as body-mass index (BMI) of 30.0 or higher.”

Regarding the long-term data, we have browsed the database again and indeed, there have been few improperly inserted denominators, which have been corrected as presented in the manuscript in Table 3. We hope that after corrections, the manuscript has seen an improvement as far as the Reviewer is concerned.

Remark #7
:
In the multivariable analysis, Mallampati IV has been found as an independent predictor of active oxygen therapy (OR: 3.089, 95% CI 1.654-5.770), but not of all-cause mortality at 60 days, as presented in Figure 2.” These were the most important findings; however, the authors did not clearly describe the details. (1) Did you mean Mallampati I as baseline for comparison for AOT? If yes, you should also show the result of other grades (II, III). Why did you only show platelet count and lung involvement (not significant data)? How about other parameter? (2) What is the baseline (also Mallampati I?) for comparison for 60-day mortality? What is the data for Mallampati II, III, and IV? Why did you only show significant parameters (leukocytes, age, lung involvement, and CKD)? How about other parameter?

Response #7:
We would like to explain the information presented in the Figure, as we believe that the information derived from the multivariable analysis are of crucial importance for the entire manuscript.
The initial manuscript Figure has been designed to present only data, which were significant in the multivariable analysis. The Mallampati II and III grades did not meet statistical significance, therefore they were not presented in the Figure. However, for the more appropriate understanding of the message conveyed by the Manuscript we have decided to modify the Figure and attach the results of multivariable analysis for Mallampati II and III grades.

All other parameter, such as those mentioned by the Reviewer did not meet statistical significance, therefore, they were not presented in the Figure. The factors included in the initial Univariable analysis, after the exclusion of co-dependent variables in the correlation analysis, are presented in the Table attached below, which has also been used in Response #2.

For the all-cause mortality at 60 days, Mallampati score did not significantly influence the Results, and therefore has not been presented in the Figure. The Figure B shows only the variables that were independently associated with increased risk of death, both regarding the categorical and continuous variables.

Therefore, in order to clarify the nature of the Figure, we have decided to modify the Title of the Figure 2, which now states Figure 2: Multivariable analysis indicating independent predictors and the influence of consecutive Mallampati grades on the need for active oxygen therapy during hospitalization (A) and independent predictors of all-cause mortality at 60 days (B)”

Round 2

Reviewer 1 Report

Dear authors, thanks for your reply.

I would mention in the limitations the lack of the ABG results and the reasons.

In the discussion I would mention the low vaccination rate and their severity of disease correlated risks because, despite vaccination has not been demonstrated to be a significant predictor of either active oxygen therapy necessity, or of mortality among your study patients, it has been largely demonstrated in the literature therefore it might be related to the small size of the population included in your study. 

 thank you

Author Response

We are very thankful for the Remarks stated during the review process for the Journal of Clinical Medicine, in which the further areas of the article entitled “Mallampati score is an independent predictor of active oxygen therapy in patients with COVID-19”, which required improvement were identified. As in the first round of the Review, we have addressed the Remarks in the „Remark-Response” formula presented below:

Reviewer #1:

Remark #1:

I would mention in the limitations the lack of the ABG results and the reasons.

Response #1:
We would like to thank the Reviewer for this comment.  We have added the following phrase to the manuscript limitation part:

„Third, the decision to obtain arterial blood gas analysis on admission was solely at the discretion of the physician in-charge, therefore in the majority of patients, the results of arterial blood gas analysis were not available for the purpose of this analysis. Nevertheless, it should be noted that in patients in whom the arterial blood gas analysis was performed, especially on the days of massive admissions, and in patients in worse clinical presentation, the passive oxygen treatment could have been implemented before arterial blood gas assessment was performed. Thus, the generalization of the results could have been significantly biased by the passive oxygen supply from the emergency medical services.”

Remark #2:
In the discussion I would mention the low vaccination rate and their severity of disease correlated risks because, despite vaccination has not been demonstrated to be a significant predictor of either active oxygen therapy necessity, or of mortality among your study patients, it has been largely demonstrated in the literature therefore it might be related to the small size of the population included in your study. 

Response #2:
We agree with the Reviewer that the issue of vaccination rate definitely requires attention. As stated before, the rate of vaccination did at least partially reflect the overall low vaccination rate in Poland, and therefore, the worse epidemiological situation of the country during the 3rd and the 4th waves of the pandemic.

As we do share the point of view of the Reviewer, we have added the following phrase of the manuscript discussion part:

„Additionally, an important factor influencing the high mortality rate of studied patients is a small percentage of full vaccination, which was 31.2% in the overall studied cohort. Although in our analysis, vaccination has not been found to independently increase the risk of either active oxygen therapy or mortality, it may be due to the relatively small studied population, and the presence of other confounders, especially since contemporary large studies indicate a significant impact of lack of full vaccination on worse prognosis and higher mortality in patients with COVID-19(37,38).”

Moreover, two new References have been added as 37 and 38:

[37] Tenforde MW, Self WH, Adams K, Gaglani M, Ginde AA, McNeal T, et al. Association Between mRNA Vaccination and COVID-19 Hospitalization and Disease Severity. JAMA [Internet]. 2021;326(20):2043–54. Available from: http://www.ncbi.nlm.nih.gov/pubmed/34734975

[38] Lopez Bernal J, Andrews N, Gower C, Robertson C, Stowe J, Tessier E, et al. Effectiveness of the Pfizer-BioNTech and Oxford-AstraZeneca vaccines on covid-19 related symptoms, hospital admissions, and mortality in older adults in England: test negative case-control study. BMJ [Internet]. 2021;373:n1088. Available from: http://www.ncbi.nlm.nih.gov/pubmed/33985964

Reviewer 2 Report

The manuscript has much improvement.

Author Response

We would like to thank the Reviewer for the appreciation of the contents of the Manuscript, along with the acceptance of the modifications made to fulfil the expectations of both Reviewers.